

# Sequencing and analyses on chloroplast genomes of *Tetrataenium candicans* and two allies give new insights on structural variants, DNA barcoding and phylogeny in Apiaceae subfamily Apioideae

Lu Kang[1], Dengfeng Xie[1], Qunying Xiao[2], Chang Peng[1], Yan Yu[1] and Xingjin He[1]

[1] Key Laboratory of Bio-Resources and Eco-Environment of Ministry of Education, College of Life Sciences, Sichuan University, Chengdu, Sichuan, people's republic of China
[2] College of Ecological Engineering, Guizhou University of Engineering Science, Bijie, Guizhou, China

## ABSTRACT

**Background**. *Tetrataenium candicans* is a traditional Chinese folk herbal medicine used in the treatment of asthma and rheumatic arthritis. Alongside several Tordyliinae species with fleshy roots, it is also regarded as a substitute for a Chinese material medicine called 'Danggui'. However, a lack of sufficient sampling and genomic information has impeded species identification and the protection of wild resources.

**Methods**. The complete chloroplast genomes of *T. candicans* from two populations, *Tetrataenium yunnanense* and *Semenovia transilliensis*, were assembled from two pipelines using data generated from next generation sequencing (NGS). Pseudogenes, inverted repeats (IRs) and hyper-variable regions were located by Geneious 11.1.5. Repeat motifs were searched using MISA and REPuter. DNA polymorphism and segment screening were processed by DNAsp5, and PCR product was sequenced with Sanger's sequencing method. Phylogeny was inferred by MEGA 7.0 and PhyML 3.0.

**Results**. The complete chloroplast genomes of *T. candicans* from two populations, *T. yunnanense* and *S. transilliensis*, were 142,261 bp, 141,985 bp, 142,714 bp and 142,145 bp in length, respectively, indicating conservative genome structures and gene categories. We observed duplications of *trnH* and *psbA* caused by exceptional contractions and expansions of the IR regions when comparing the four chloroplast genomes with previously published data. Analyses on DNA polymorphism located 29 candidate cp DNA barcodes for the authentication of 'Danggui' counterfeits. Meanwhile, 34 hyper-variable markers were also located by the five Tordyliinae chloroplast genomes, and 11 of them were screened for population genetics *of T. candicans* based on plastome information from two individuals. The screening results indicated that populations of *T.candicans* may have expanded. Phylogeny inference on Apiaceae species by CDS sequences showed most lineages were well clustered, but the five Tordyliinae species failed to recover as a monophyletic group, and the phylogenetic relationship between tribe Coriandreae, tribe Selineae, subtribe Tordyliinae and *Sinodielsia* clade remains unclear.

Corresponding author
Xingjin He, xjhe@scu.edu.cn

**Discussion**. The four chloroplast genomes offer valuable information for further research on species identification, cp genome structure, population demography and phylogeny in Apiaceae subfamily Apioideae.

## INTRODUCTION

Subtribe Tordyliinae is an Apiaceae lineage in the tribe Tordylieae whose species are characterized by enlarged radiant outer petals and dorsally compressed fruits, with more than 15 incorporated genera (*Fu, 1981*; *Xiao, 2017*). *Tetrataenium candicans* var. *candicans* (Wall. ex DC.) Manden. is a perennial Tordyliinae herb endemic to the Qinghai-Tibet Plateau (QTP) and adjacent regions. They typically grow on sunny slopes and forest edges at elevations of about 2,000 to 4,200 m, with dense white pubescence on the back side of leaves and prominent fruit ridges distinguishing them from other allied species (*Fu, 1981*). Our survey of the Chinese Materia Medica resource inventory in Sichuan, Yunnan, Tibet and Qinghai province showed that *T. candicans* was a common herbal medicine used to treat asthma and rheumatic arthritis. Moreover, alongside some Tordyliinae species with fleshy roots, it has been also widely used as a substitute for 'Danggui,' a traditional Chinese medicine that was first recorded in ''Shen Nong's Herbal Classic'' in about AD 200 to be used in blood enrichment and the treatment of pulmonary diseases. However, since the misuse of the active ingredients in this species might cause severe side effects (*Sondhiaa et al., 2017*; *Zhang, Wang & Hu, 2001*), the *Chinese Pharmacopoeia Commission (2015)* has restricted 'Danggui's' origin plant to *Angelica sinensis* (Oliv.) Diels only. Additionally, the long growth cycle and over-exploitation of wild individuals used for medicinal and economic purposes has menaced the existence of *T. candicans* (*Joshi & Dhar, 2003*). Less attention was given to discriminate among the counterfeit 'Danggui' plants (especially from the subtribe Tordyliinae), which also delayed research on the drug's safety and discovery.

DNA barcoding is a common technique that aims to standardize DNA segments for species-level discrimination, and has been widely used in identification, taxonomy and biodiversity studies (*Hodgetts et al., 2016*; *Chen et al., 2009*; *Devloo-Delva et al., 2016*). The general procedure in developing DNA barcodes involves building public libraries of DNA segments from known species. The target sequences from an unknown species are then matched up with barcode libraries to search for best-matching species or close relatives (*Hajibabaei et al., 2007*). With the decline of sequencing cost in the past decades, a vast amount of DNA barcodes has been uploaded and retrieved. For instance, the *CO1* gene is applied in the species recognition of animals and fungus (*Seifert et al., 2007*), while ITS, ITS2, *psbA-trnH*, *matK* and *rbcL* are effective in identifying Apiaceae species (*Liu et al., 2014*). Nevertheless, due to different substitution rates, hybridization, multiple copies and mutations, DNA barcodes are not applicable to all species (*Kress & Erickson, 2007*). For example, a universal barcode *ycf15* (*Gao, Zhao & Ni, 2017*) is absent in *Coriandrum*

*sativum* (*Peery, 2015*), and species discrimination by ITS sequences are often flanked by hybridization (*Liu et al., 2011*). For these reasons, it is essential to develop specialized available DNA markers to protect and fully utilize local herbal resources.

The chloroplast (cp) is an important component in the plant cell as it is where optical energy can be restored in the form of carbohydrates through photosynthesis. As a semi-autonomous organelle, the chloroplast possesses uni-parental inherited cricoid DNA with a length ranging from 20.98 kb (*Sciaphila densiflora*) to 1320.6 kb (*Haematococcus lacustris*), presenting conservative gene locations and categories in flowering plants (*Molina et al., 2014*). Because of these features, chloroplast genomes have expanded our knowledge of genetic engineering (*Daniell et al., 2016*), phylogeny (*Jansen et al., 2006*) and population demography. With the development of next generation sequencing (NGS) and assembly technologies, the past two decades have witnessed more than 100 cp genomes reportedly covering over 40 genera in Apiaceae. These cp genomes are all characterized by a quadripartite structure similar to those of most angiosperms, and their discrepancies are mainly reflected in rare changes in gene order and shifting IR-LSC boundaries (*Peery, 2015*; *Spooner et al., 2017*). Although it is a monophyletic group (*Logacheva et al., 2010*) that is of important economic value, few studies have been devoted to the chloroplast genomes of Tordyliinae species.

In the present study, we reported cp genomes of two *T. candicans* individuals, *Tetrataenium yunnanense* (*Xiao et al., 2017*) and *Semenovia transiliensis*. By comparing them with published cp genomes, we explored the structural variation among Apiaceae species and located hypervariable cp DNA markers for Tordyliinae species and candidate cpDNA barcodes for the species identification of 'Danggui' counterfeits. Additionally, based on two cp genomes of *T. candicans*, 11 cpDNA markers were screened to survey population diversity. Finally, phylogeny on 37 Apiaceae and Araliaceae species using CDS sequences of cp genomes was also inferred. This research will offer genomic and genetic information for future phylogenetic and phylogeographic studies on Tordyliinae, and will shed light on the protection and utilization of wild herbal medicine resources.

## MATERIALS & METHODS

### Plant materials and DNA sequencing

Mature and healthy *T. candicans* leaves from two populations, *S. transiliensis* and *T. yunnanense*, were collected from Ganzi (Sichuan province, China; coordinates: 31°63′N, 100°01′E; approval number: XQY20150814001), Mu li (Sichuan province, China; coordinates: 28°40′N, 101°01′E; approval number: KL20180620001), Tekesi (Xinjiang province, China; coordinates: 42°86′N, 81°84′E; approval number: XQY20160724008) and Lanping (Yunnan province, China; coordinates: 26°07′N, 99°85′E; approval number: KL20180802001), respectively. All voucher specimens were deposited in the Sichuan University Herbarium (SZ). The total genomic DNA was isolated from dry leaves using an improved CTAB method (*Doyle, 1987*) and sequenced at Novogene (Novogene BioTech, Inc. Beijing, China) by Illumina Hiseq 2500 platform (Illumina, San Diego, CA). A genome skimming sequencing strategy (*Steven, 2015*) was performed to obtain deep coverage
of organelle genomes regardless of the shallow sequencing of total genomic DNA. The coverage map was generated by Geneious 11.1.5.

## Quality control, genome assembly and genome annotation

The quality of raw data was assessed by FastQC (v0.11.7 for windows) (*Andrew, 2014*). To accommodate the demands for data preparation of assembly software, we merely removed primers and adapters using Cutadapt v1.1.8 (*Martin, 2011*), and the qualified data was then assembled in different ways. Initially, a pipeline combining bowtie2-build (*Langmead & Salzberg, 2012*), SAMtools (*Li et al., 2009*), BEDtools (*Quinlan, 2014*) and SOAPdenovo 2 (*Luo et al., 2012*) was used to pick up reads that mapped to the best reference cp sequence. The consensus sequences were generated by Geneious 11.1.5 (*Kearse et al., 2012*) and gaps were filled by Sanger sequencing. For comparison, a seed-based assembler named NOVOPlasty 2.7.2 (*Dierckxsens, Mardulyn & Smits, 2017*) was employed which eventually cyclized cp genomes.

New rules for annotating cp genomes of the four assembled cp genomes were also followed: (1) Geneious 11.1.5 was used for batch annotation of consensus sequences referring to annotations of *Daucus carota* (NC_008325) and *Pastinaca pimpinellifolia* (NC_027450); (2) Sequences were re-annotated on the DOGMA (*Wyman, Jansen & Boore, 2004*) website to check for any omissions and the identity was set above 70% in protein-coding genes, 80% in rRNA genes and 90% in tRNA genes; (3) Pseudogenes were not marked, and incomplete homologous segments of a real gene (such as *psbA*, *ycf1*) were also regarded as pseudogenes; (4) tRNAscan-SE v2.0 (*Lowe & Chan, 2016*) was used for verifying tRNA genes. All annotation errors were manually corrected. Finally, the circular map was generated by OGDRAW (*Lohse et al., 2013*).

## Repeat motifs

MISA (*Thiel, 2003*) was used for identifying simple sequence repeats (SSRs) of *T. candicans*, *T. yunnanense*, *Heracleum moellendorffii*, *S. transiliensis* and *P. pimpinellifolia*. The threshold was set as follows: the minimum repetitions of SSRs for mono-nucleotide, di-nucleotides, tri-nucleotides, tetra-nucleotides, penta-nucleotide and hexa-nucleotides should be 10, 5, 4, 3, 3 and 3, respectively. Simple dispersed repeats (SDRs) were identified by Reputer (*Kurtz et al., 2001*). The minimal size of SDRs was 30 bp, and the similarity among SDRs should be no less than 90%. IRA region and double-counting SDRs were removed.

## DNA polymorphism and candidate DNA barcodes

To identify candidate DNA barcodes for the discrimination of 'Danggui' and local counterfeits, the cp genomes of *T. candicans, T.yunnanense* and five Selineae species (*Angelica acutiloba*) (NC_029391) *Angelica decursiva* (KT781591), *Angelica gigas* (KX118044), *Angelica laxifoliata* (NC_040122) and *A. sinensis* (MH430891) were employed. DNA polymorphism of hyper-variable regions was calculated by DnaSP5 (*Librado & Rozas, 2009*). Before that, cpDNA segments were aligned by MAFFT v7.419 (*Katoh et al., 2002*) and trimmed manually. The genome variation was visualized using mVISTA (*Frazer et al., 2004*) by the Shuffle-LAGAN alignment program (*Brudno et al., 2003*). The haplotype

analysis, Tajima's test, and Fu and Li's test were processed by DNAsp5. Additionally, the DNA polymorphism of cp genomes of five Tordyliinae species (*T. candicans*, *T. yunnanense*, *Heracleum moellendorffii*, *S. transiliensis* and *P. pimpinellifolia*) was also calculated.

## Segment screening for *T. candicans*

To investigate the population diversity of *T. candicans* in order to protect wild resources, we screened 11 cpDNA segments from noncoding regions and designed primers using Geneious 11.1.5 based on hyper-variable regions of the two cp genomes of *T. candicans*. More than 30 populations were sampled covering most distributions on record, and each population sampled one or two individuals. Total genomic DNA was extracted by an improved CTAB method (*Doyle, 1987*). PCR amplification was carried out in a 30 μL volume, and included 15 μL ddH$_2$O, 9 μL mix (Tiangen, Beijing, China), 3 μL DNA solution, 1.5 μL Forward primer and 1.5 μL reverse primer solution (10 μmol/L$^{-1}$). The PCR procedure was as follows: pre-denaturation for 4 min at 94 °C, followed by 35 cycles of denaturation for 45 s at 94 °C, annealing at 50–55 °C for 1 min and extension at 72 °C for 1 min, and finally, extension at 72 °C for 7 min. The PCR products were sequenced (BGI, Beijing, China) by ABI 310 Genetic Analyzer (Applied Biosystems, Waltham, MA, USA). The paired-end sequences were assembled by Geneious 11.1.5. DnaSP5 was available for nucleotide diversity, haplotype diversity, and mismatching analyses.

## Phylogeny reconstruction

Phylogenetic analysis was performed using concatenated alignments of 80 protein coding sequences from 37 plastomes: *A. sinensis* (MH430891), *Anethum graveolens* (KR011055), *Angelica decursiva* (KT781591), *Angelica gigas* (KX118044), *Angelica laxifoliata* (NC_041022), *Angelica polymorpha* (NC_041580), *Anthriscus cerefolium* (GU456628), *Arracacia xanthorrhiza* (KY117235), *Bupleurum latissimum* (KT983258), *Caucalis platycarpos* (KX832334), *Chuanminshen violaceum* (KU921430), *Cicuta virosa* (NC_037711), *Cnidium officinale* (MH121055), *Coriandrum sativum* (NC029850), *Crithmum maritimum* (HM596072), *Daucus carota* (NC_008325), *Dendropanax morbifer* (KR136270), *Foeniculum vulgare* (KR011054), *Glehnia littoralis* (KT153022), *H. moellendorffii* (MK210561), *Hansenia forbesii* (KX808492), *Ledebouriella seseloides* (KT153021), *Ligusticum sinense* (KX594382), *Ligusticum tenuissimum* (KT963039), *Ostericum grosseserratum* (KT852844), *P. pimpinellifolia* (NC_027450), *Petroselinum crispum* (HM596073), *Peucedanum japonicum* (KU866530), *Pimpinella rhomboidea* var. *tenuiloba* (MG719855), *Pleurospermum camtschaticum* (KU041142), *Prangos trifida* (MG386251), *Semenovia gyirongensis* (NC_042912), *Seseli montanum* (KM035851), *S. transiliensis* (MN267864), *T. candicans* (MK333395), *Tiedemannia filiformis* subsp. *greenmannii* (HM596071), *T. yunnanense* (MN365275). *Dendropanax morbifer* (KR136270) and *Bupleurum latissimum* (KT983258) were treated as outgroup. CDS sequences were extracted by Geneious 11.1.5 and aligned using the MAFFT plugin, with a proper manual cutoff by MEGA 7.0 (*Kumar, Stecher & Tamura, 2016*). To avoid Long Branch Attraction (LBA) (*Huelsenbeck, 1997*), Maximum likelihood (ML) estimation was performed for phylogeny inference by online tool phyML 3.0 (*Guindon et al., 2010*) with

1,000 bootstrap replicates, and the best-fit model was GTR+G+I. Non-conserved loci were filtered by Gblocks 0.91 b (*Talavera & Castresana, 2007*) with default parameters.

## RESULTS

### Genome features of *T. candicans*, *S. transiliensis* and *T. yunnanense*

The sequences of two *T. candicans*, *S. transiliensis* and *T. yunnanense*, were 142,261 bp, 141,948 bp, 142,145 bp and 142,714 bp in length, respectively, presenting typical quadripartite structures containing a large single copy region (LSC) and a small single copy region (SSC) jointed by two inverted repeats (IRA and IRB) (Fig. 1, coverage map in Fig. S1). The overall GC content of the four cp genomes was 37.4% with an exception of 37.3% in *T. yunnanense*. The GC content of the IR region was much higher than that of the other components, mainly because tRNA and rRNA genes had aggregated at this region. The SSC region had the lowest GC content in comparison with the remaining two components (Table 1).

The cp genomes of *T. candicans*, *T. yunnanense* and *S. transiliensis* contain 123 genes (Table 2): 80 protein-coding genes, 35 tRNA genes and eight rRNA genes. There was no protein-coding gene but five tRNA genes (*trnA-UGC*, *trnH-GUG*, *trnI-GAU*, *trnN-GUU*, *trnR-ACG*) and four rRNA genes (*rrn4.5*, *rrn5*, *rrn16*, *rrn23*) were doubled in the IR regions. The SSC region included 11 protein-coding genes (*ccsA*, *ndhA*, *ndhD*, *ndhE*, *ndhF*, *ndhG*, *ndhH*, *ndhI*, *rpl32*, *rps15*, *psaC*) and one tRNA gene (*trnL-UAG*), while the remaining 68 protein-coding genes and 24 tRNA genes occurred at the LSC region, with *ycf1* significantly straddling the SSC and IRA regions. There were 11 protein-coding genes (*atpF*, *ndhA*, *ndhB*, *petB*, *petD*, *rpl16*, *rpl2*, *rpoC1*, *rps12*, *rps16*, *ycf3*) and six tRNA genes (*trnA-UGC*, *trnG-UCC*, *trnI-GAU*, *trnK-UUU*, *trnL-UAA*, *trnV-UAC*) that had introns, most of which had only one intron except for *clpP*, *rps12* and *ycf3*. Coding regions of *rps12* were located across the LSC and two IR regions.

### Gene loss and pseudogenization

Pseudogenes (marked with 'Ψ') were DNA segments that were homologous to real genes but had lost functionality to some extent (*Vanin, 1985*). Of the four assembled chloroplast genomes, ψ *ycf15* in *T. candicans*, *T. yunnanense* and *S. transiliensis* was formed through several small InDels (KU951523 was used as a reference sequence), while ψ *ycf1* and ψ *psbA* were generated through LSC-IR boundary shifting.

### LSC-IR/IR-SSC boundaries

Taking the IRA-LSC boundary of tobacco as reference, junction type I', defined by Downie (*Plunkett & Downie, 2000*), was found in cp genomes of *T. yunnanense, T. candicans* and *S. transiliensis.* Meanwhile, a rare LSC-IR junction named F' was discovered in chloroplast genomes of *P. pimpinellifolia* (NC_027450) and *H. moellendorffii* (MK210561). The two junction types experienced extra expansion to *psbA,* and the duplicated segment *trnH-psbA* inserted right behind old contracted LSC-IRB boundaries (Fig. 2).

While boundaries between the IR and LSC region were quite changeable, IRB-SSC and SSC-IRA junctions were highly conserved. All IRB-SSC boundaries of the four chloroplast

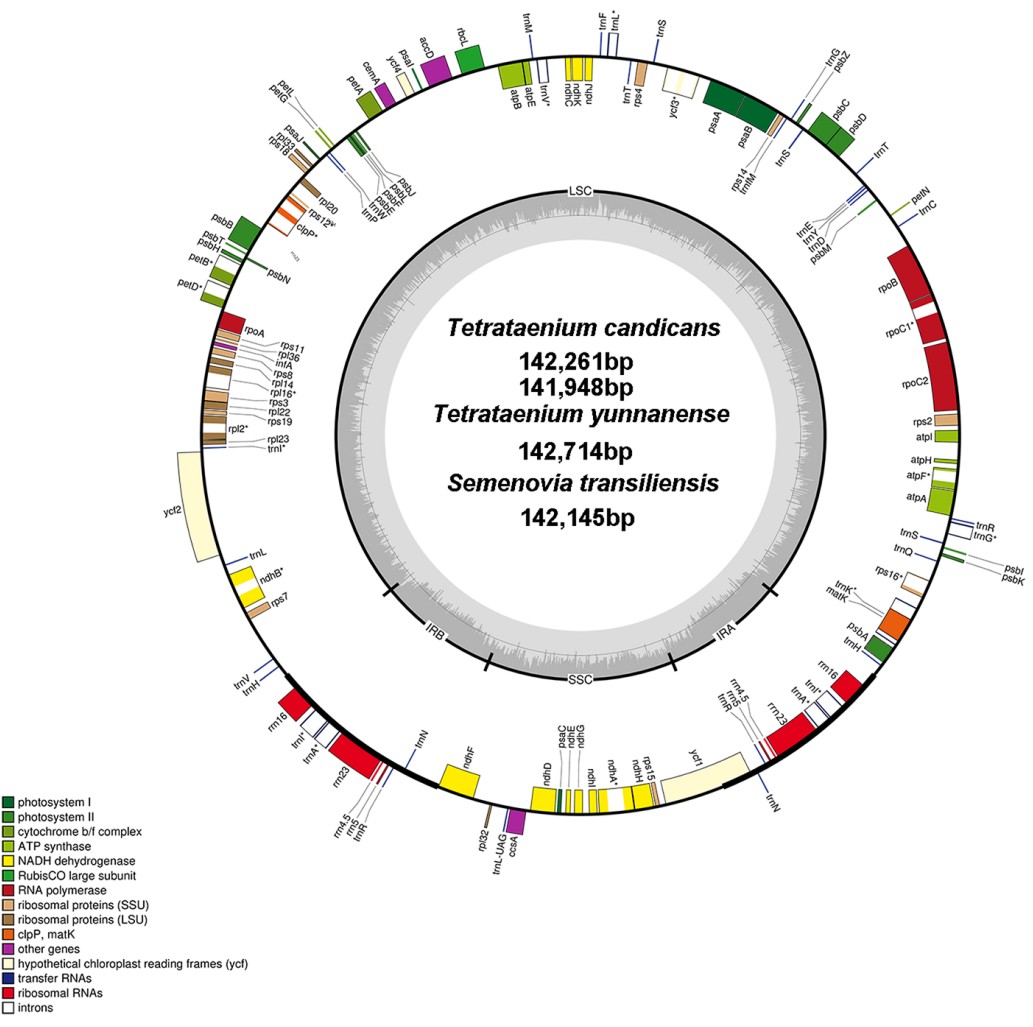

- photosystem I
- photosystem II
- cytochrome b/f complex
- ATP synthase
- NADH dehydrogenase
- RubisCO large subunit
- RNA polymerase
- ribosomal proteins (SSU)
- ribosomal proteins (LSU)
- clpP, matK
- other genes
- hypothetical chloroplast reading frames (ycf)
- transfer RNAs
- ribosomal RNAs
- introns

**Figure 1** **Chloroplast genome map of *T. candicans*, *S. transiliensis* and *T. yunnanense*.** Genes outside the circle are transcribed in a counterclockwise direction, while genes inside are transcribed in a clockwise direction. The darker gray belt in the inner circle refers to the GC content, and the lighter gray color refers to the AT content. Genes that have introns are marked with (*). SSC, small single copy; LSC, large single copy; IRA and IRB, Inverted repeats A and B.

genomes reported here were placed adjoined to *ndhF*, and the SSC-IRA boundaries located within the scope of *ycf1*. The Junction type definitions will contribute to further research on structural variations of cp genomes of Apiaceae species.

## SSRs and SDRs

SSRs (Simple sequence repeats) correlated with multiple replications of motifs that contained one to six base pairs. Herein, we reported SSRs in cp genomes of the five Tordylinae species that were longer than 10 bp (Fig. 3, Supplemental Information 1). Overall, 58, 64, 54, 52 and 70 SSRs were detected in *T. candicans* (MK333395), *T. yunnanense*, *H. moellendorffii*, *P. pimpinellifolia* and *S. transiliensis*, respectively (Figs. 3A–3E). 'A/T'-rich mono-nucleotide SSRs were the most abundant microsatellites in

Table 1 **Sampling and genome information of four assembled cp genomes.** The average coverage of each cp genomes was calculated by formula: (mapped reads * 150)/length of cp genome.

| | S. transiliensis | T. candicans | T. candicans | T. yunnanense |
|---|---|---|---|---|
| Genebank No. | MN267864 | MK333395 | MK522402 | MN365275 |
| Length of Genome(bp) | 142,143 | 142,261 | 141,948 | 142,714 |
| Mean coverage (X) | 1805 | 3371 | 187 | 908 |
| Raw data (GB) | 17.6 | 16.3 | 5.1 | 26.6 |
| GC content (%) | 37.4 | 37.4 | 37.4 | 37.3 |
| LSC length (bp) | 100,072 | 99,569 | 99,961 | 100,000 |
| LSC GC content (%) | 36.1 | 36.1 | 36.1 | 36.0 |
| SSC length (bp) | 17,513 | 17,536 | 17,533 | 17,514 |
| SSC GC content (%) | 31.1 | 31.1 | 31.1 | 30.9 |
| IR length (bp) | 12,279 | 12,578 | 12,227 | 12,600 |
| IR GC content (%) | 47.1 | 47.1 | 47.1 | 46.8 |
| CDS length(bp) | 68,586 | 68,577 | 68,561 | 65,044 |
| CDS GC content (%) | 37.8 | 37.9 | 37.9 | 37.9 |
| Protein coding genes | 80 | 80 | 80 | 80 |
| tRNAs | 35 | 35 | 35 | 35 |
| rRNAs | 8 | 8 | 8 | 8 |
| Total genes | 123 | 123 | 123 | 123 |

all cp genomes, followed by 'AT/AT' di-nucleotide SSRs, while 'G/C'-rich SSRs were scarce. In *T. candicans*, 79.3% (46) of SSRs were polyA, polyT or poly AT, and the proportions changed into 79.7% (51), 88.7% (47), 78.8% (41) and 81.4% (57) in *T. yunnanense*, *H. moellendorffii*, *P. pimpinellifolia* and *S. transiliensis*, respectively. Analyses on the distribution of different categories of SSRs in these five species suggested that the LSC region contains more SSRs than the SSC and IR regions. Repetitions of SSRs (Fig. 3E) in each cp genome were substantially in agreement and larger repeat units meant fewer replications. These SSRs can be potential markers for species discrimination, phylogeny and population studies for the five Tordyliinae species.

SDRs (Short Dispersed Repeats) were more complex repeats longer than 30bp. In this study, 24, 42, 27, 25, and 20 pairs of SDRs in cp genomes of *T. candicans* (MK333395), *T. yunnanense, H. moellendorffii, P. pimpinellifolia* and *S. transiliensis* were surveyed, respectively (Fig. S2, Supplemental Information 3). Forward (direct) SDRs were the most frequent repeats, followed by palindromic SDRs and reverse SDRs. Most SDRs were located at the intergenic regions; *trnH-rrn16* specifically was a hotspot area for SDRs in *T. yunnanense* with 15 pairs of SDRs dispersed in this region. There was no SDR longer than 60 bp in *H. moellendorffii,* nor *P. pimpinellifolia,* while only two, two, and three SDRs longer than 60 bp were observed in *T. candicans, S. transiliensis* and *T. yunnanense*, respectively. SDRs longer than 100 bp only existed in *T. yunnanense*.

Additionally, SDRs accounting for the expansion of cp genomes through multiple replications were found in *S. transiliensis* and *T. yunnanense*. The 39 bp SDRs that began at the 30419th genome in *S. transiliensis* repeated three times in the LSC region. In *T. yunnanense,* SDRs that repeated more than twice occurred in the IR and LSC regions, with

**Table 2  Genes and categories in the four assembled Tordyliinae cp genomes.** Doubled genes are marked with asterisks (*).

| Groups | Categories | Name of genes | | | | | |
|---|---|---|---|---|---|---|---|
| | rRNAs | rrn4.5* | rrn5* | rrn16* | rrn23* | | |
| | tRNAs | trnY-GUA | trnW-CCA | trnV-UAC | trnV-GAC | trnT-UGU | trnT-GGU |
| | | trnS-UGA | trnS-GGA | trnS-GCU | trnR-UCU | trnR-ACG* | trnQ-UUG |
| | | trnP-UGG | trnN-GUU* | trnM-CAU* | trnL-UAG | trnL-UAA | trnL-CAA |
| | | trnK-UUU | trnI-GAU* | trnI-CAU | trnH-GUG* | trnG-UCC* | trnF-GAA |
| | | trnE-UUC | trnD-GUC | | | | |
| Self-replication | Small subunit of ribosome | rps2 | rps3 | rps4 | rps7 | rps8 | rps11 |
| | | rps12 | rps14 | rps15 | rps16 | rps18 | rps19 |
| | Large subunit of ribosome | rpl36 | rpl33 | rpl32 | rpl23 | rpl22 | rpl20 |
| | | rpl16 | rpl14 | rpl2 | | | |
| | RNA polymerase subunits | rpoA | rpoB | rpoC1 | rpoC2 | | |
| | Subunits of photosystem I | psaA | psaB | psaC | psaI | psaJ | |
| | Subunits of photosystem II | psbZ | psbT | psbN | psbM | psbL | psbK |
| | | psbJ | psbI | psbH | psbF | psbE | psbD |
| | | psbC | psbB | psbA | | | |
| | Subunits of cytochrome | petN | petL | petG | petD | petB | petA |
| | Subunits of ATP synthase | atpI | atpH | atpF | atpE | atpB | atpA |
| Photosynthesis | Large subunit of Rubisco | rbcL | | | | | |
| | Subunits of NADH-Dehydrogenase | ndhK | ndhJ | ndhI | ndhH | ndhG | ndhF |
| | | ndhE | ndhE | ndhD | ndhC | ndhB | ndhA |
| | Maturase | matK | | | | | |
| | Envelope membrane protein | cemA | | | | | |
| | Subunit of acetyl-CoA | accD | | | | | |
| | C-type cytochromesynthesis gene | ccsA | | | | | |
| | Protease | clpP | | | | | |
| Other genes | open reading frames | ycf1 | ycf2 | ycf3 | ycf4 | | |
| | Translational initiation factor | infA | | | | | |

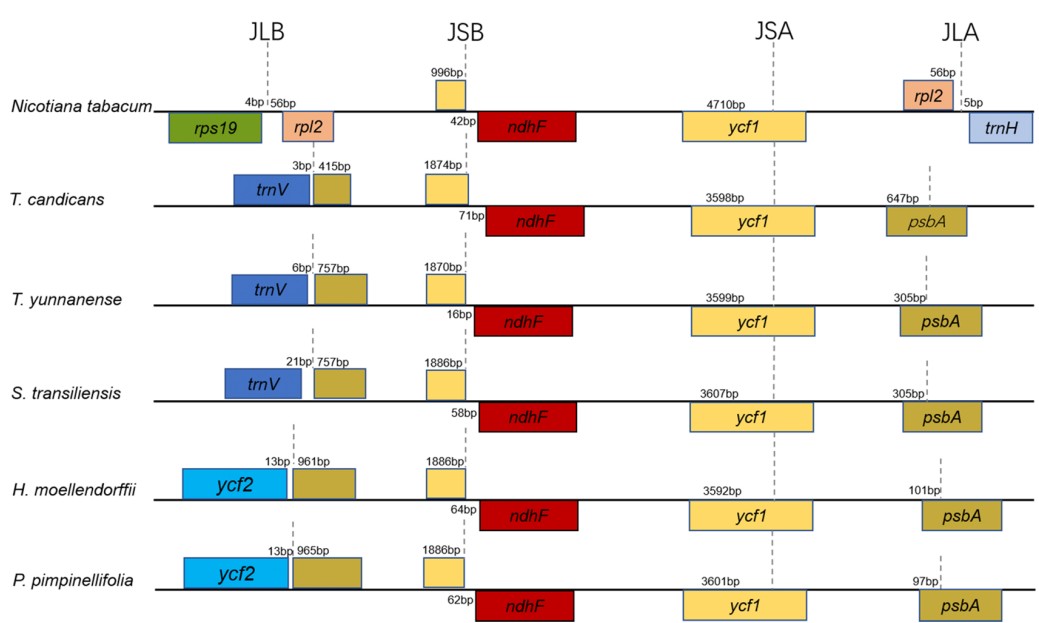

**Figure 2 Comparison of the borders of the LSC, SSC, and IR regions of the chloroplast genomes of the five Tordyliinae species and *Nicotiana tabacum*.** Genes beyond black lines are transcribed from right to left, genes over the black lines—from left to right. Each colored block represents a gene, and same color means same gene. The smaller blocks with the same color are truncated genes caused by shifting of IR boundaries. JLA, junction of LSC and IRA regions; JLB, junction of LSC and IRB regions; JSA, junction of SSC and IRA regions; JSB, junction of SSC and IRB regions.

lengths of 30 bp (six replications), 32 bp (five replications) and 31 bp (three replications). Particularly, the 125 bp SDRs were considered novel insertions as they aligned any species with high similarities. This information on SDRs will be advantageous in exploring the structure evolution of chloroplast genomes.

## DNA polymorphism

The visualization of cp genomes of seven species by mVISTA (Fig. 4) indicated that most hyper-variable DNA segments were situated at intergenic regions and introns, while CDS regions were much more conservative. DNA polymorphism of 67 hyper-variable regions were calculated with Pi values ranging from 0.00306 (*psbB*) to 0.03549 (*trnH-psbA*), with 38 sharing nucleotide diversity beyond 0.01, and 29 DNA segments able to discriminate each species (by number of haplotypes, Table S1). Beyond that, we also observed long insertions in three DNA segments (*psbA*, *ndhF-rpl32* and *ndhC-trnV*) exclusively existing in cp genomes of *A. sinense*. The 28 DNA segments (*accD-ycf4, atpH-atpI, ccsA-ndhD, clpP intron2, intron, matK, ndhA ndhC-trnV, ndhF-rpl32, psbA-matK, psbE-petL, psbK-psbI, psbZ-rps14, rpl32-trnL, rpoB-trnC, rpoC1 intron, rpoC2-rpoC1, rps16 intron, rps16-trnQ, trnC-petN, trnE-trnT, trnG intron, trnH-psbA, trnK-rps16, trnL-ndhJ, trnS-trnG, trnT-psbD, trnT-trnL, trnV-atpE*) and segments with unique insertions could be candidate plastid DNA barcodes used in the authentication of counterfeit 'Danggui'.

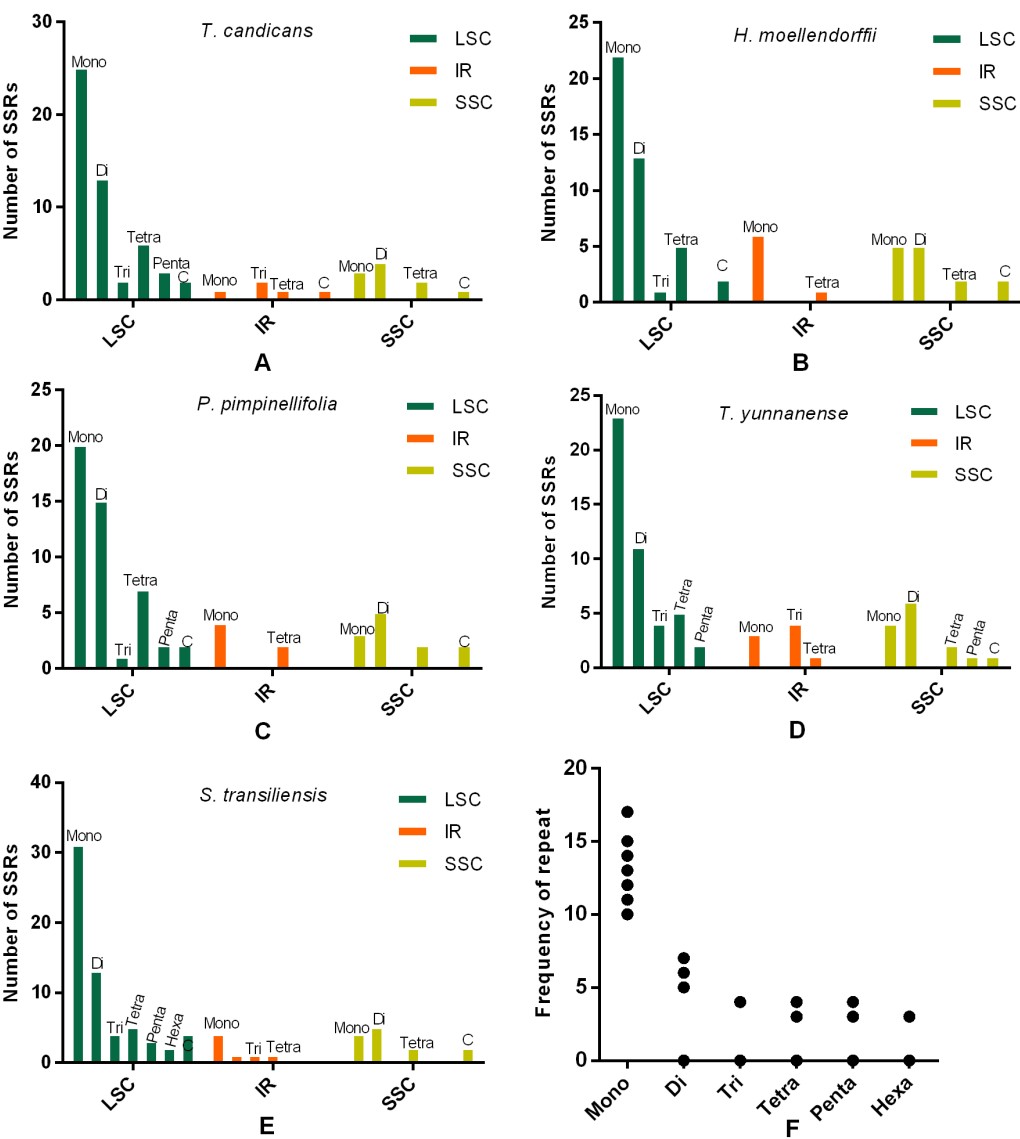

**Figure 3** **Numbes and repeat frequency of SSRs in five Tordyliinae species.** (A–E) Number of SSRs in LSC, SSC and IR regions in plastomes of the five Tordyliinae species. (F) Repeat frequency of different kinds of SSRs. Type C SSRs that blended more than one SSR type are calculated separately. Mono, mono-nucleotide; Di, di-nucleotides; Tri, tri-nucleotides; Tetra, tetra-nucleotides; Penta, penta-nucleotides; Hexa, hexa-nucleotides; C, compound SSRs.

DNA polymorphism of 106 intergenic regions and 19 introns, as well as 77 CDS in cp genomes of five Tordyliinae species, was also calculated. The nucleotide diversity value (pi) in CDS regions (Fig. S3) ranged from 0 to 0.013 with a mean of 0.0048, which was much lower than that of intergenic regions (from 0 to 0.0681, with a mean of 0.014) and introns (from 0 to 0.0142, with a mean of 0.009). *matK* exhibited the highest DNA polymorphism in the CDS region with 42 polymorphic (segregating) loci detected. Of the 106 non-coding regions (Fig. 5), 65 DNA segments had DNA polymorphism values that

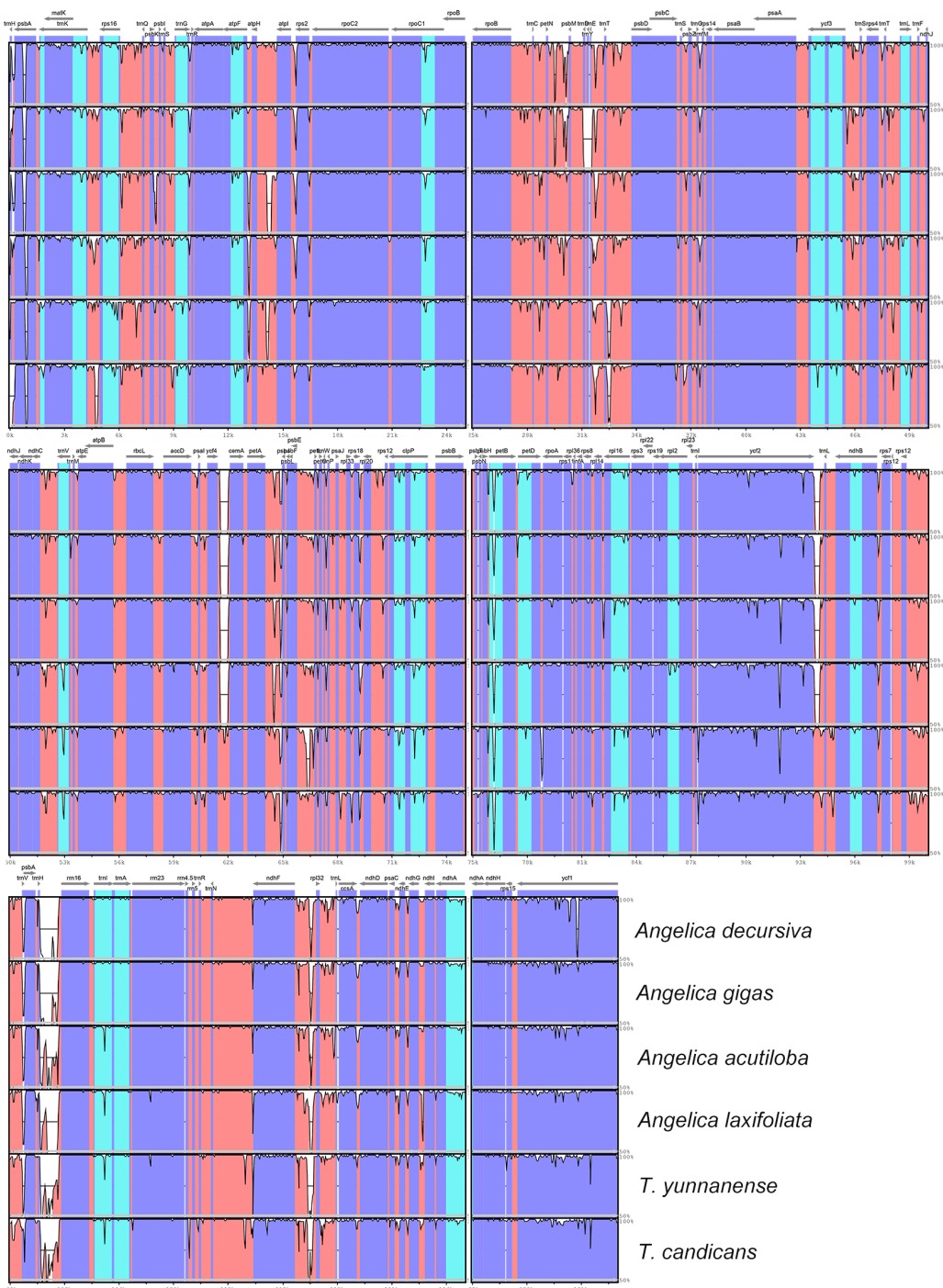

**Figure 4** **mVISTA visualization of alignment of cp genomes of *Angelica sinensis* and other six species.** Blue blocks represent introns, orange blocks represent intergenic regions, and purple blocks represent exons. The locations of genes in cp genomes are shown below blocks, and sequence identity is shown on the right of blocks. Genes are marked with grey arrows, and arrows in forward direction represent genes are transcribed from left to right, otherwise from right to left.

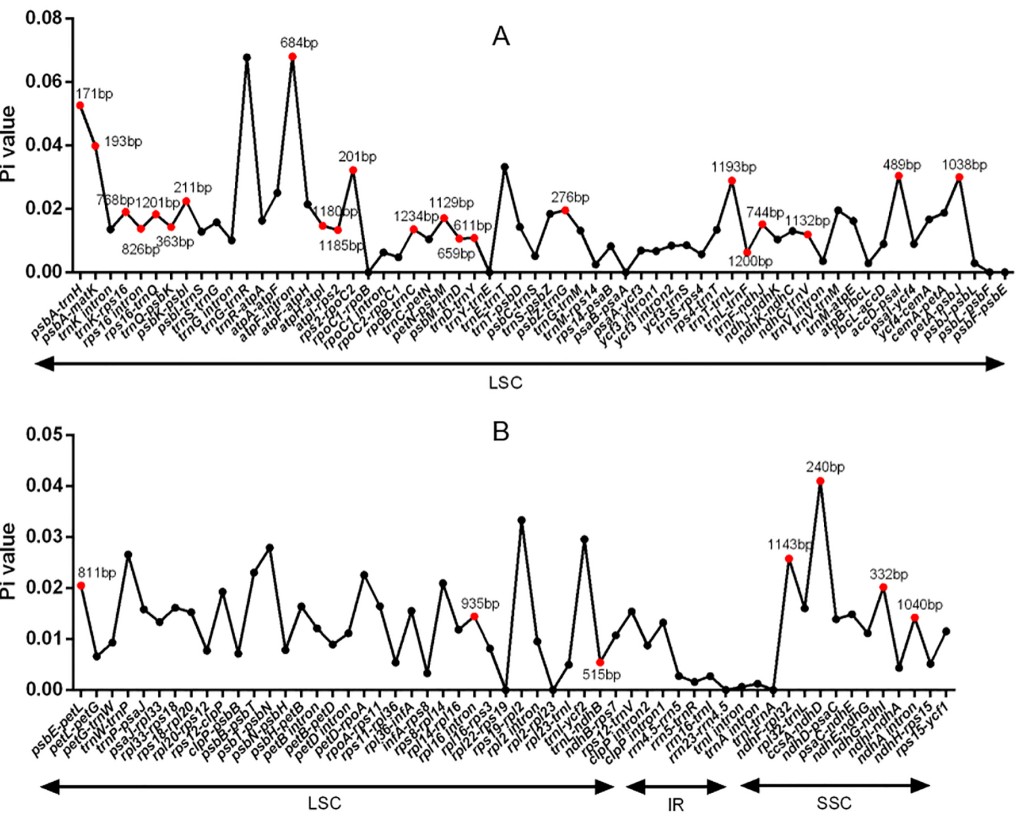

**Figure 5** **DNA polymorphism of non-coding regions of chloroplast genomes of the five Tordyliinae species.** (A) DNA polymprphism of non-coding segments at LSC (large single copy). (B) DNA polymorphism of non-coding segments at LSC, SSC (small single copy) and IR (inverted repeat) regions. Segments that are marked with length and red dots are applicable for DNA markers. Pi value, nucleotide diversity value.

exceeded 0.01, with 42 segments longer than 200 bp. Only seven introns had relatively high DNA polymorphism that exceeded 0.01. Finally, according to length and number of polymorphism sites, 29 segments of non-coding regions (*accD-psaI, atpF intron, atpH-atpI, atpI-rps2, ccsA-ndhD, ndhA intron, ndhC-trnV, ndhF-rpl32, ndhG-ndhI, petA-psbJ, petN-psbM, psbA-matK, psbA-trnH, psbE-petL, psbK-psbI, psbM-trnD, psbZ-trG, rpl16 intron, rpoB-trnC, rps16 intron, rps16-trnQ, rps2-rpoC2, trnD-trnY, trnF-ndhJ, trnK-rps16, trnL-ndhB, trnL-trnF, trnQ-psbK, trnT-trnL*) and five CDS (*ccsA, matK, ndhI, ndhG* and *rps14*) regions were selected to be potential DNA markers used for further research.

## Segments screening for genetic diversity of *T. candicans*

All target DNA segments were successfully amplified and sequenced, except for sequencing failures in *ndhF-rpl32* and *rpl16* intron caused by repeat sequences. These segments from different populations presented low DNA polymorphism below 0.01 (Table S2), but significantly high haplotype diversity from 0.248 (*trnS-trnG*) to 0.844 (*rpl16 intron*). Only four haplotypes were found in *trnS-trnG* and *rpl16* intron alignments, while *trnQ-rps16* had the most haplotypes with 16. More than half of the haplotypes were private, indicating high

haplotype diversity. Regarding mismatch analyses, all nine segments presented unimodal or smooth curves, indicating that the sampled populations (Table S3) had probably experienced recent expansion. However, the results were only confirmed by Tajima's test and Fu and Li's test against *rps16 intron*, *trnQ-rps16*, *trnL-trnT*, *rps16-trnK* (weak statistical significance), and *trnS-trnG* (weak statistical significance), while test result from *rpl16 intron*, *psbA-trnH*, *trnL-F* did not show statistical significance of deviation from zero.

## Phylogenetic inference

Maximum Likelihood (ML) trees (Fig. 6) by 80 concatenated CDS sequences of 37 species from Apiaceae subfamily Apioideae were constructed, and most lineages were well supported. Bupleureae species experienced the earliest differentiation in the 12 clades, followed by species from tribe Pleurospermeae, *Komarovia* clade, tribe Oenantheae and tribe Scandiaceae. Seven clades or tribes from Apioid superclade clustered together with 100% bootstrap support value, and Pyramidoptereae was sister to the remaining six clades. Within the subtribe Tordyllinae, lineage including *P. pimpinellifolia* and *H. morllendorffii* were closely clustered with 100% bootstrap support value. This lineage, together with species from tribe Selineae and *Sinodielsia* clade, constituted three parallel branches. Lineage composed of *T. candicans*, *S. transiliensis*, *T. yunnanense* and *C. sativum* was clustered with species from *Sinodielsia* clade with 100% bootstrap support value. Phylogeny inference failed to recover the five Tordyliinae species as a monophyletic group.

# DISCUSSION

## Assembly methods and gene content

In this study, four chloroplast genomes were assembled using genome skimming data based on the principle that organelle DNA have more copies than nrDNA in plant cells. The cp genomes of *T. candicans*, *S. transiliensis* and *T. yunnanense* were quadripartite with conservative gene organization and an almost halved length of IR regions that determined the selection of assembly methods. From experience, seed-based assemblers (such as NOVOPlasty v 2.7.1) were more competent in assembling chloroplast genomes with contracted IR regions.

The four cp genomes contained 123 genes: 80 protein-coding genes, 35 tRNA genes and eight rRNA genes. Apart from truncated *psbA*, only one pseudogene, $\psi$ *ycf15*, was observed, suggesting that gene content was highly conserved. 'ATG' was the start codon of most plastid genes, and illegal start codons occurred in *ndhD*, *psbC* and *rps19* in the four reported cp genomes. The translational initiation sites of *ndhD* was 'ACG', which might be associated with a post-transcriptional 'C' to 'U' edit to improve the translating efficiency (*Hirose et al., 1999*). The start codon of *psbC* was annotated as two distinct codons, namely 'GTG' in tobacco and 'ATG' in Apiaceae species, resulting from annotation preference as this gene was highly conservative in both tobacco and the four cp genomes, and 'ATG' was 33 bp upstream of 'GTG'. 'ATG' was reported as a putative start codon of *rps19* in algae, while in angiosperms, the start codon turned into 'GTG', indicating an ancient mutation that may take place in a common ancestor.

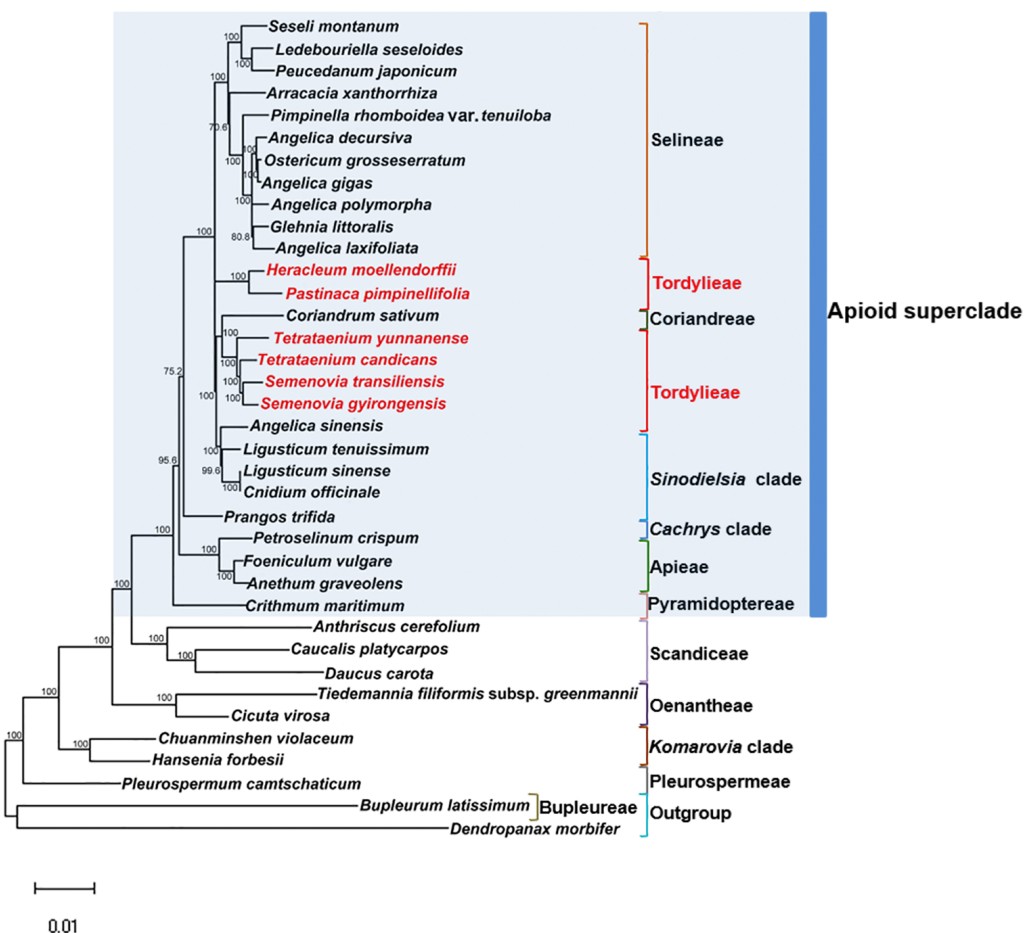

**Figure 6** **Maximum likehihood (ML) tree inferred from 80 concatenated protein coding sequences of 37 plastomes of Apiaceae and Araliaceae species.** Bootstrap support values higher than 50% are indicated. The scale bar corresponds to 0.01 substitutions per site. Different clades or tribes are marked with colored line segments. The blue bar in the right of ML tree and the grey block represent species from Apioid superclade. Species from tribe Tordylieae are colored red.

## Molecular markers and segment screening for population genetics

Based on the number of haplotypes and insertions, the 29 candidate cpDNA barcodes were able to distinguish 'Danggui' from the counterfeits, which gave new insight on discovering new candidate DNA barcodes using cp genomes. The 34 hypervariable DNA segments and SSRs in the five Tordyliinae species were also located. Our findings were consistent with previous conclusions that most SSRs were abundant in 'A/T' format, but the number of SSRs in the five cp genomes was much less than what had been reported in other species (*Zhao et al., 2018*; *Liu et al., 2018*; *Zong et al., 2019*).

Additionally, we made attempts on segments screening for the genetic diversity of *T. candicans* through two cp genomes. All nine DNA segments exhibited high haplotype diversity with abundant private haplotypes, which might be explained by habitat fragmentation (*Fahrig, 2003*) of *T. candicans* in the Qinghai-Tibet plateau and adjacent

regions, while low DNA polymorphism might be correlated to low substitution rate of cpDNA segment or rapid adaptive evolution (*Zhao et al., 2013*; *Shi & Zhang, 2015*). What's more, recent expansion shown by mismatch analyses might give insight into why *T. candicans* are more likely to survive and be prosperous in its habitats. Those segments of high haplotype diversity could be genetic resources for population genetics and complement nrDNA barcodes in identifying producing areas of *T. candicans*.

## Dynamic IRA-LSC boundaries and repeat sequences

Most angiosperms were highly conserved in the cp genome structure. As described in *Nicotiana tabacum*, a typical LSC–IRB ($J_{LB}$) boundary occurred near *rps19*. However, due to contraction or expansion at the IR regions, LSC-IRB or IRA-LSC boundaries could be really dynamic in Apiaceae. So far, 12 kinds of IRA-LSC boundaries were defined in the family Apiaceae (*Plunkett & Downie, 2000*; *Peery, 2015*), among which type I' and F' were two rare junction types among land plants which existed only in cp genomes of tribe Coriandreae and Tordyliinae species in family Apiaceae.

There are several reported explanations for changes in the IR-LSC boundaries, including homologous recombination (*Guo et al., 2014*), double-strand break repair and the existence of SDRs (*Odom et al., 2008*). A previous study on cp genomes of *C. sativum* supported double-strand break repair (DBSR) by partial duplication of *trnV-GAC* and SDRs located within IR regions that caused change in the IR boundary (*Peery, 2015*). However, the duplication of *trnV-GAC* was not observed in any Tordyliinae species. More interestingly, we observed some direct SDRs located between *trnH-GUG* and *trnL-CAA*, and between *trnH-GUG* and *rrn16* in cp genomes of the five Tordyliinae species. These results indicated that changes in the LSC- IR boundary were far more complicated to be explained by a single mechanism. Additionally, some of the aforementioned SDRs (i.e., 125 bp SDRs in *T. yunnanense*) are new insertions that are highly similar to nrDNA, mtDNA or mRNA segments. The novel insertions existed in the form of direct SDRs that might be preliminary evidence for intra-molecular recombination and foreign DNA transformation mediated by short direct-repeat sequences (*Ogihara, Terachi & Sasakuma, 1988*; *Cai et al., 2008*).

## Phylogeny inference

In the past decades, phylogeny on the whole range of Apiaceae or subfamily Apioideae has been widely explored (*Downie, Downie & Watson, 2000*; *Downie et al., 2004*; *Downie et al., 2010*; *Zhou et al., 2008*; *Zhou et al., 2009*), but few attempts have been made on utilizing plastid genomes for phylogeny inference. In such cases, protein coding sequences from 37 plastomes from 12 tribes in Apiaceae were employed for phylogeny inference as they were maternal inheritance, free of hybridization and had more informative loci (*Daniell et al., 2016*).

Reconstructed ML tree had 12 well stated tribes. The phylogenetic relationship of most tribes was in congruence with previous phylogenetic inferences using nrDNA datasets (*Downie et al., 2010*; *Zhou et al., 2008*; *Zhou et al., 2009*). *Ostericum koreanum* (KT852844, marked as *Ostericum grosseserratum* in NCBI database), revised as *Angelica reflexa* (*Lee et al., 2013*), was distant from tribe Oenantheae, and the placement of *L. tenuissimum*

and *L. sinense* in *Sinodielsia* clade by cp phylogenomics was also proven in Downie's research (*Downie et al., 2010*). Within subtribe Tordyliinae, *T. candicans, T. yunnanense* and *S. transiliensis* clustered as a lineage, while *H. moellendorffii* was closely related to *P. pimpinellifolia*, which supported the conclusion that both *Pastinaca* L. and *Heracleum* L. should be accepted into *Heracleum* sensu stricto clade (*Logacheva et al., 2010*; *Xiao, 2017*).

However, the five species from subtribe Tordylinae were not recovered as a monophyletic group on ML tree as studies using ITS and ETS sequences suggested (*Downie et al., 2010*; *Zhou et al., 2008*; *Xiao, 2017*; *Logacheva et al., 2010*). The possible reasons for phylogenetic incongruency were sampling errors, systematic errors and biological factors (*Zou & Ge, 2008*). Sampling errors (stochastic errors) and systematic errors were excluded first as causes for topological discordances as abundant informative loci existed in chloroplast genomes and no conspicuous long branch attraction (*Bergsten, 2005*) was observed (*Philippe et al., 2005*; *Zhang, Zeng & Li, 2012*). Biological factors referred to many aspects, such as horizontal gene transfer, ancient hybridization, incomplete lineage sorting, homoplasy and concert evolution (*Koch, Dobes & Thomas, 2003*). Even though very few intergeneric hybrids had been reported on Apiaceae species (*Desjardins et al., 2015*; *Yu et al., 2011*), we suggested ancient hybridization cannot be excluded as most Tordyliinae and Selineae species possessed the same base number of chromosomes ($n = 11$) (*He, Pu & Wang, 1994*). Moreover, as shown in Fig. 6, short branches suggested that the disputed phylogenetic relationship may also be explained by incomplete lineage sorting in Apiaceae superclade Apioid that may had experienced rapid evolutionary radiation and species formation (*Calviño, Martínez & Downie, 2008*; *Tamura et al., 2012*) related to climate fluctuations or reproductive isolation during Miocene (*Liao et al., 2012*; *Wu et al., 2014*; *Banasiak et al., 2013*). Nevertheless, further analyses with more nuclear markers and samples from Tordylieae should be performed in order to clarify this issue. Our research also indicated that chloroplast genomes were effective in phylogeny inference, but may not be competent in dealing with conflicting phylogeny among rapidly radiated species.

## CONCLUSIONS

In this study, four cp genomes of two *T. candicans* individuals, *T. yunnanense* and *S. transiliensis*, were first reported. Analyses on genome structure revealed two kinds of rare simultaneous contractions and expansions of LSC-IR boundaries in assembled Tordyliinae cp genomes. The candidate cpDNA barcodes for the authentication of 'Danggui' and 34 hypervariable DNA segments in the five Tordyliinae cp genomes were also identified. Segment screening for population genetics of *T. candicans* suggested that populations had probably experienced recent expansion. Phylogeny inferences based on protein coding sequences from 37 plastomes of Apiaceae and Araliaceae species suggested that subtribe Tordyliinae was closely related with subtribe Selineae, tribe Coriandreae and *Sinodielsias* clade. However, the ML tree failed to recover the five Tordyliinae species as a monophyletic group. On that basis, forthcoming research might focus on the structure variation of plastomes of Apiaceae species, cpDNA barcodes for 'Danggui' and phylogeny reconstruction at the genomic level by exploring efficient methods and increasing samples.

 

## ACKNOWLEDGEMENTS

The authors thank the researchers who made advancements on the phylogeny of Apiaceae, and the publishers of related cp genome sequences in NCBI.

### Funding

This work was supported by the National Natural Science Foundation of China (Grant Nos. 31872647, 31570198), and the Chinese Ministry of Science and Technology through the 'National Science and Technology Infrastructure Platform' project (Grant No. 2005DKA21403-JK). The funders had no role in study design, data collection and analysis, decision to publish, or preparation of the manuscript.

### Grant Disclosures

The following grant information was disclosed by the authors:
National Natural Science Foundation of China: 31872647, 31570198.
Chinese Ministry of Science and Technology: 2005DKA21403-JK.

### Competing Interests

The authors declare there are no competing interests.

### Author Contributions

- Lu Kang conceived and designed the experiments, performed the experiments, analyzed the data, prepared figures and/or tables, authored or reviewed drafts of the paper, approved the final draft.
- Dengfeng Xie and Xingjin He conceived and designed the experiments, authored or reviewed drafts of the paper, approved the final draft.
- Qunying Xiao authored or reviewed drafts of the paper, approved the final draft, sample collection.
- Chang Peng analyzed the data, authored or reviewed drafts of the paper, approved the final draft.
- Yan Yu contributed reagents/materials/analysis tools, authored or reviewed drafts of the paper, approved the final draft.

### Field Study Permissions

The following information was supplied relating to field study approvals (i.e., approving body and any reference numbers):

Field experiments were approved by the National Specimen Information Infrastructure (NSII) (XQY20150814001, KL20180620001, XQY20160724008, KL20180802001).

### DNA Deposition

The following information was supplied regarding the deposition of DNA sequences:

The chloroplast sequences are available in the Supplemental Files and at GenBank: MK333395, MN267864, MN365275 and MK522402.

PeerJ ______________________________________

## Data Availability

   The sequenced segments are available in the Supplemental Files.

## Supplemental Information

Supplemental information for this article can be found online at http://dx.doi.org/10.7717/peerj.8063#supplemental-information.

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
