# Peer review of "Sequencing and analyses on chloroplast genomes of Tetrataenium candicans and two allies give new insights on structural variants, DNA barcoding and phylogeny in Apiaceae subfamily Apioideae"

_PeerJ, doi:10.7717/peerj.8063_

## Round 0.1 · original submission · Major Revisions

I decided to give you an opportunity to improve your article although the comments of the reviewers are numerous and important. Please necessarily address the following issues: 1) send the paper for review to a company or to an English editor, the paper does not reach a professional English level, include the name of the company/editor in the next version of the manuscript, 2) there are many issues in methods and results with pseudogenization, corroborate the data downloaded from GenBank before using it, 3) clarify your results compared with these downloads, 4) validate aligment of your data matrix to be certain that pseudogenes are not a problem of aligment, 4) confirm whether contractions and expansions of populations are not an effect of the previous mentioned problems.

Reviewer 1 ·

Basic reporting

English needs correction, errors and mistypings are present.

Experimental design

no coment

Validity of the findings

no comment

Additional comments

This study by Lu Kang and coauthors presents a very welcome and quite interesting plastid genomes of 4 representatives of the Apiaceae family. To my opinion, the authors obtained novel results of high interest for readers of PeerJ especially those working on molecular evolution of organelle genomes, plant barcoding and phylogeny.

Although I am very positive about this paper, it can not be accepted for publication in its current form and requires appropriate revisions. The main issue I see is the way the authors deal with plastome pseudogenes.

As follows from figure 2, Lu Kang et al have found ycf1 pseudogenization in Tetrataenium yunnanense plastome. This is very surprising as it is a very rare event, as far as I know ycf1 pseudogenization was reported only in some Passifloraceae, Fabaceae and Geraniaceae (putting aside monocots). Apiaceae ycf1 sequence alignment shows an insertion of 101 bases, 100 of which perfectly match with adjacent sequence, and this duplication looks like just an assembly error. However, I did not find in the manuscript any attempts to confirm the presence of pseudo-ycf1 in T. yunnanense by read mapping and PCR amplification and sequencing. If they were made, this fact should be reported in the manuscript, if they were not - they should be done.

Arracacia plastome hardly contains pseudo-rpoC1, because when annotation starts with the next Methionine downstream (see, for example, Prunus plastome KF990036), pseudo-rpoC1 will become uninterrupted and functional. This is just a matter of annotation, not pseudogenization.

Lu Kang and coauthors describe and discuss also other pseudogenes in Apiaceae plastomes. But everything retrieved from GeneBank should be checked twice, especially when the accession status is "unpublished". To my opinion, most of the discussed "pseudogenes" should be treated as putative unless confirmed by Sanger sequencing. The most abundant in putative pseudogenes plastome sequence of Carum carvi was obtained with pyrosequencing technology, the most drawback of which is problematic reading of mononucleotide stretches (see, for example, Tsuruta et al 2017, DOI:10.1371/journal.pone.0169992). When a stretch of 9A (instead of 10A) causes a frameshift, this is a doubtful frameshift and needs to be confirmed, but Peery (2015) did not report and, by the way, did not discuss anything concerning plastid pseudogenes! I wonder why, and I wonder why the authors believe that rpoB and rpoC2 (subunits of the plastid encoded RNA-polymerase, which transcribes photosynthesis-related genes) can be pseudogenized in non-parasitic/non-heterotrophic plant without checking it by themselves.

And the last, ycf68 and orf42 are extremely rare annotated (pseudo)genes. The authors also did not annotate them and did not describe them, giving the readers an opportunity to waste their time trying to find out what are the authors talking about. As for ycf15, currently two different ycf15 genes can be found in GeneBank plastome annotations: "real"-ycf15 directly after ycf2 and "fake"-ycf15 somewhere between rps12_3`end and trnV-GAC, what is more, in some accessions both are annotated as ycf15 (for example, Camellia japonica KU951523). The former is really pseudogenized, but Selineae members plastomes contain at least first 40 bases, and in Coriandrum full-length pseudo-ycf15 can be found, while the authors treat all of them "absent".

Summing up, this part of the manuscript should be rewritten or deleted, as it may require additional experimental work. Pseudogenes representing truncated copies of genes are just worth mentioning, as they arise due to the IR border shift, which is discussed properly in corresponding section.

Besides, I make a few comments as follows.
In Materials and Methods information about number of contigs and their length should be added, as well as stringency parameters used in GBlocks. As I mentioned above, reads mapping on the assembled genome sequences is necessary.

In Results the authors describe their own results and normally any references should be avoided, therefore text on lines 270-275 and 312-316 can be transferred to Discussion.
Line 449: " caused trnH(GUG)or psbA inversions" - not inversions, but duplications due to IR expansion at JLA.

Figures and Tables:
Figure 5 - unclear meaning of "P,R,C,F"
Table 2 - "unknown function" should be deleted, as ycf3 and ycf4 are photosystem assembly proteins. Just "open reading frames" is enough.

Reviewer 2 ·

Basic reporting

The manuscript includes results of the research of four chloroplast genomes and phylogeny in Apiaceae subfamily Apioideae. By comparing the four chloroplast genomes with previous published data, universal 59 pseudogenization and shifting IR-LSC boundaries in Apiaceae were recognized. The manuscript has huge data, supported with figures and tables. The style of the presentation of the results in manuscript is simple and understandable, but a bit chaotic.
I have following remarks:
1. In 283 type D shift is recalled for two Prangos species. Chloroplast genomes of three Prangos Lindl. species were published (MG386251, KY652265, KY652266) and D shift is recognized in all three species, that is why it will be correct to mention 3 Prangos species in the manuscript. Also, it will be interesting if two other Prangos cp genomes are included in phylogenic analyses of Apiaceae species.
2. In 334 the novel insertion of 125bp is mentioned. It will be interesting to find if these sequences are matching to the sequences from other genomes (such as nuclear or mitochondrial).
3. The manuscript has technical incorrections such as absence of gaps between the text and parentheses, absence of commas and others.
4. Both Figure and Table legends should be expanded with details which help readers to understand their meaning.

Experimental design

no commen

Validity of the findings

no comment

Additional comments

In general, the manuscript is highly interesting from scientific point and results will be useful in further researches on species identiļ¬cation, cp genome structure, population demography and phylogeny in Apiaceae subfamily Apioideae. I recommend this manuscript for publication after making corrections mostly of technical character.

Reviewer 3 ·

Basic reporting

This manuscript was relatively well organized in the introduction, but authors messed the results and discussions up, which are main problems must be well reorganized before acceptance. The authors must strictly stick to only describe results they found basing on their own analyses. And leaving all comparisons between their findings and previous publications as well as description of results in previous publications to discussion if they were necessary. The secondly main problem is that shared CDS genes in plastomes of all Apiaceae species should be used for building phylogeny and performing following analyses instead of complete plastome, especially lots of pseudogenizations existed in the Apiaceae plastomes. Besides, add titles or necessary descriptions to the supplementary files.

Experimental design

Experiments in this manuscript were well designed, which used adequate techniques.

Validity of the findings

The main problems are that authors described too much findings basing on previous publications instead of their own data, which have been emphasised in above. Secondly, the "34 DNA segments and SSRs for phylogenic and phylogeographic studies" that were emphasised as part of main results were not clearly described and presented in the manuscript and supplementary files. Thirdly, "ubiquitous pesudogenization and two kinds of rare simultaneous contractions and expansion in five Tordyliinae species" were not really existed in comparing the number of pesudogenization in the four newly sequenced plastomes with other species in the same family.

Additional comments

Authors must change the results and discussion in way like I have emphasised above before final acceptance decision.

Annotated reviews are not available for download in order to protect the identity of reviewers who chose to remain anonymous.

---

## Round 0.2 · Minor Revisions

I appreciate your effort dealing with issues raised by the two reviewers, however there are still some problems, most of them related to phylogenetic analysis which need to be presented with branch support. In addition, annotations and position of molecular markers such as psbA, trnH, ycf1, ndhF, rps19 and adjacent rpl2 are transcribed only in one direction. The rest of comments are included below.

Reviewer 1 ·

Basic reporting

The authors have made many improvements, and the current manuscript is of a much higher quality than the first version. The authors have adequately addressed my comments, but there are points, however, which make me feel that the manuscript is still not in a condition where it is ready for publication: figures and references. Correction of the text is still needed ("monophynic", "Selieae", "37 complete CDS sequences", "natural group", etc)

Experimental design

Experiments well designed, appropriate methods used

Validity of the findings

no comment

Additional comments

I make several comments as follows.

1. The authors report results of newly performed phylogenetic analyses. Conclusions are the same, but I can not find the rationale for presenting a single ML tree without branch support values (Figure 6), while the authors declared ML analysis with bootstrap resampling. A "small" tree (Figure 7) is redundant, because it presents nothing new about non-monophyletic Tordylieae and nothing about tribes relationship, as the latter depends on root placement. So I suggest to add to the "big" tree bootstrap support values (higher than 50%) and remove Figure 7 with subsequent manuscript correction.
2. Figure 3: junctions indication is wrong and should be corrected. Some genes are shown also wrongly, because genes psbA, trnH, ycf1, ndhF, rps19 and adjacent rpl2 are transcribed in one direction, see my comments in annotated manuscript.
3. Legends to figures 1, 3 and 6 should be changed, I added suggestions in annotated manuscript.
4. Citations in the text do not correspond to the Reference list, I found 7 references not cited in the text and 3 citations not included in References.
5. "Phylogeny reconstruction" section of Materials & Methods contains a list of plants, which corresponds to their position in a phylogenetic tree, but looks like disordered. Why do not the authors arrange plants in alphabetical order (see, for example https://doi.org/10.7717/peerj.7830)? Genbank accession numbers for newly assembled plastomes also can be provided here.
6. Correction of the text is still needed ("monophynic", "Selieae", "37 complete CDS sequences", "natural group", etc)

Annotated reviews are not available for download in order to protect the identity of reviewers who chose to remain anonymous.

Reviewer 3 ·

Basic reporting

The revised manuscript is basically ok now.

Experimental design

No more comments of experimental design.

Validity of the findings

But I still find that the annotations of gene rps12 and introns in the Figure 1 were wrong. Because the gene rps12 obviously cannot cover such large range in the plastome, which the present figure shows. Besides, introns are also not displayed in the Figure 1.

---

## Round 0.3 · accepted · Accept

Thank you for considering suggestions by the two reviewers and myself. In this version, I only found some typographical errors in some of the figure legends, such like in Figure 5. I suggest that you careful read the proof version to correct these problems.